# Characterization of MYBL1 Gene in Triple-Negative Breast Cancers and the Genes’ Relationship to Alterations Identified at the Chromosome 8q Loci

**DOI:** 10.3390/ijms25052539

**Published:** 2024-02-22

**Authors:** Audrey Player, Sierra Cunningham, Deshai Philio, Renata Roy, Cydney Haynes, Christopher Dixon, Lataja Thirston, Fawaz Ibikunle, Taylor Allen Boswell, Ayah Alnakhalah, Juan Contreras, Myra Bell, Treveon McGuffery, Sahia Bryant, Chidinma Nganya, Samuel Kanu

**Affiliations:** 1Department of Biology, Texas Southern University, Houston, TX 77004, USA; sierracu@usc.edu (S.C.); d.philio3155@student.tsu.edu (D.P.); renataproy@gmail.com (R.R.); cydney.haynes@yahoo.com (C.H.); l.thirston8917@student.tsu.edu (L.T.); ibikunlefawaz@gmail.com (F.I.); t.allenboswell9040@student.tsu.edu (T.A.B.); ayah.alnakhalah@gmail.com (A.A.); j.contreras3273@student.tsu.edu (J.C.); m.bell8614@student.tsu.edu (M.B.); t.mcguffery6204@student.tsu.edu (T.M.); s.bryant6199@student.tsu.edu (S.B.); c.nganya2651@student.tsu.edu (C.N.); s.kanu4742@student.tsu.edu (S.K.); 2Department of Environmental and Interdisciplinary Sciences, Texas Southern University, Houston, TX 77004, USA; c.dixon3789@student.tsu.edu

**Keywords:** triple-negative breast cancer (TNBC), v-Myb Avian Myeloblastosis Viral Oncogene Homolog-Like 1 (MYBL1), Valosin Containing Protein Interacting Protein 1 (VCPIP1), c-MYC Proto-Oncogene, BHLH Transcription Factor (MYC), Block Of Proliferation 1 Protein (BOP1)

## Abstract

The MYBL1 gene is a strong transcriptional activator involved in events associated with cancer progression. Previous data show MYBL1 overexpressed in triple-negative breast cancer (TNBC). There are two parts to this study related to further characterizing the MYBL1 gene. We start by characterizing MYBL1 reference sequence variants and isoforms. The results of this study will help in future experiments in the event there is a need to characterize functional variants and isoforms of the gene. In part two, we identify and validate expression and gene-related alterations of MYBL1, VCIP1, MYC and BOP1 genes in TNBC cell lines and patient samples selected from the Breast Invasive Carcinoma TCGA 2015 dataset available at cBioPortal.org. The four genes are located at chromosomal regions 8q13.1 to 8q.24.3 loci, regions previously identified as demonstrating a high percentage of alterations in breast cancer. We identify alterations, including changes in expression, deletions, amplifications and fusions in MYBL1, VCPIP1, BOP1 and MYC genes in many of the same patients, suggesting the panel of genes is involved in coordinated activity in patients. We propose that MYBL1, VCPIP1, MYC and BOP1 collectively be considered as genes associated with the chromosome 8q loci that potentially play a role in TNBC pathogenesis.

## 1. Introduction

The MYBL1 gene is a member of the MYB family of genes, which includes c-MYB, MYBL1 and MYBL2. The genes are proto-oncogenes and strong transcriptional activators that play a role in cell survival, proliferation, and differentiation, all events that are key to cancer progression [1]. Data showed that a loss-of-function of the c-MYB gene in adult hematopoietic stem cells led to impaired proliferation and accelerated differentiation [2,3], and truncations in the gene led to oncogenesis [4]. The MYBL2 gene regulates cell cycle progression, cell differentiation and survival [5], as does MYBL1, which is also a master regulator of male meiosis [6]. MYBL1 null mice are viable but have defects in spermatogenesis and breast development, implicating a role of the gene in sperm and breast cell development.

Of the MYB family members, fewer publications exist in PUBMED and at GeneCard.org characterizing the MYBL1 gene compared to scientific publications describing MYBL2 and c-MYB genes. Substantial amounts of data document the association of c-MYB and MYBL2 in a variety of cancers, including breast cancers, compared to fewer studies describing the involvement of the MYBL1 gene in cancer processes [1]. However, the trend is changing, and data documenting the MYBL1 gene’s involvement in tumor progression are increasing. Studies show MYBL1 (which is located at chromosome 8q13.1) binds to and activates the ANGPT1 gene (which is located at chromosome 8q23.1) to promote tumor angiogenesis in hepatocellular carcinoma [7]. Other studies show over-expression of MYBL1 in low-grade glioma [8], breast adenoid cystic carcinoma, which is a rare triple-negative cancer with a good prognosis [9] and tracheobronchial adenoid cystic carcinoma [10]. Wang et al. suggest the use of MYBL1 as an immunotherapy biomarker in clear cell renal cell carcinoma [11], and our data show that some TNBCs express high levels of the MYBL1 gene [12]. Related to structural changes in the gene, Fujii et al. identified genomic rearrangements and gene fusions of MYBL1 with ACTN1, AFIB and NFIB genes [13] in adenoid cystic carcinoma tumors.

Our laboratory initially identified MYBL1 gene overexpression in a subset of TNBC cell lines following meta-analyses of breast cancers processed using DNA microarrays deposited at Gene Expression Omnibus (GEO) [14]. TNBCs lack expression of estrogen receptor, progesterone receptor and HER-2 genes. The cancers are aggressive and represent 15–20% of breast cancers [15]. TNBC datasets were extracted from GEO and examined using unsupervised analysis methods. Six genes were identified as associated with TNBC compared to non-tumor and certain other breast cancer subtypes [12]. MYBL1 was one of the six genes. The fact that MYBL1 is over-expressed in TNBC and is a strong transcription factor that drives the expression of other genes implicates the gene in cancer processes.

The aim of the current study is to further characterize the human MYBL1 gene with the ultimate goal of analyzing the gene in TNBC. There are two parts to the current study. In the first part, we analyzed the MYBL1 Reference sequences (Ref Seqs) deposited at the National Center for Biotechnology Information (NCBI) [16]. We found that there are 10 unique Ref Seqs based on Multiple Sequence Alignment. NM_001080416.4 variant is the longest of the transcript variants, and it contains a unique exon 15, which corresponds to the Regulatory domain of the gene [17]. Results from the alignment studies can be useful for future analyses of the transcript variants and protein isoforms in the event alternative forms of the gene are suspected of being differentially regulated. The second part of our study involves the validation of MYBL1 and a subset of candidate genes in cell lines and a TNBC patient sample dataset. In an earlier experiment, we knocked down MYBL1 expression in a TNBC cell line and identified genes affected by the knockdown process [18]. Along with MYBL1, we identified a subset of genes located on chromosomal regions 8q13.1–8q24.3. A number of genes were identified as differentially expressed, hence affected by the knockdown process. Based on PCR and protein analyses, the most reliable candidate genes, in addition to MYBL1, were VCPIP1, MYC and BOP1 genes. In the current study, we validate RNA and protein expression of the candidate genes in additional TNBC cell lines and a patient sample dataset and examine the patient samples for concordant alterations of our gene panel. Data presented here show that a substantial number of different types of genomic and expression-related alterations in VCPIP1, MYC and BOP1 genes occur in many of the same patients as MYBL1 gene. It could be that MYBL1, VCPIP1, MYC and BOP1 genes function together to affect the genotype in certain TNBCs.

## 2. Results

### 2.1. Analyses of MYBL1 in TNBC

The current study focuses on characterizing the MYBL1 gene, with the ultimate focus on the MYBL1 gene in TNBCs. The first part of the study involves sequence analyses of the Ref Seq transcript variants and protein isoforms deposited at NCBI. The second part of the study involves analyses of the MYBL1 and several candidate genes in TNBC cell lines and patient samples. In summary, this study (a) outlines the similarities and differences between the MYBL1 transcript and isoform Ref Seqs and (b) reveals a relationship between expression and genomic alterations in MYBL1, VCPIP1, MYC and BOP1 genes located at chromosome 8q locus in cell lines and a patient sample dataset.

Part 1 of this study: Analyses of MYBL1 Reference Sequence transcripts and proteins:

### 2.2. Analyses of the Transcript Variants Deposited at NCBI

The human MYBL1 transcript variants and protein isoforms are deposited in the NCBI database as part of Gene Assembly GRCH38.p14. As of 10/02/2023, there are 17 sequences designated NCBI Ref Seqs corresponding to the MYBL1 homo sapiens gene, 10 of which are unique based on the Multalin^TM^ Sequence alignment comparison (Appendix A). The sequences include curated Ref Seqs (i.e., NM and NP designations) and predicted sequences annotated from NCBI Ref Seq contigs (i.e., XM and XP designations). The sizes of the individual transcript variants and their corresponding start sites are given in Table 1, along with the corresponding sizes of the protein isoforms. Three genes are designated NM, and all others are XM-predicted sequences. MultAlin^TM^ identifies 10 unique variant sequences, but there is a 3 nucleotide (and 1 amino acid) difference between NM2 and NM3), implying these two variants and subsequent isoforms are the same, so there are likely nine unique variants. Because NM and XM-predicted sequences are technically different based on MultAlin^TM^ analyses, all 10 of the variants are being considered for the comparisons in the current study.

The longest full-length transcript variant 1 (NM_001080416.4; NM1) has 16 exons, and the shortest NCBI Ref Seq: XM_054360507.1 (XM7) has 13 exons. The numbering of the exons is based on the longest NM_001080416.4 (NM1) variant. A diagram of the Ref Seqs is shown in Figure 1, where the percent similarity of each exon is compared to NM_001080416.4 (NM1), which is the longest. Of note, four variants contain 16 total exons, the NM_001080416.4 (NM1) and three other variants, including XM_0170013455.2 (XM1), XM_0170013456.2 (XM2) and XM_011517533.4 (XM3); unique to these variants, each contains exon 15, absent from all other variants. There is 100% similarity between the exon 15 sequences in all four variants. Alignments of the exon 15 nucleotide sequences are shown in Figure 2a. The region associated with exon 15 corresponds to the Negative Regulatory region of the NP1 MYBL1 protein [17,19]. Except for XM_054360507.1 (XM7), which contains 13 totaled exons, all other variants contain one less exon compared to NM1. The most substantial differences between the variants are (a) the presence of exon 15 in NM1, XM1, XM2 and XM3 compared to the other variants, (b) differences in the start sites and exon 1 and (c) the XM7 variant, which is the shortest variant with a total of 13 exons.

We identified MYBL1 as differentially expressed based on data (and probe sets) examined using the Affymetrix U133 plus two microarrays. The Affymetrix microarrays contain probe sets corresponding to the 3′ untranslated regions of the nucleotides. Generally, our PCR transcript probe sets are designed to correspond to regions captured by the Affymetrix microarray probe sets. The XP7 variant is truncated and does not contain this 3′ untranslated region. In the event the XM7 variant is expressed, it will not be captured by our PCR probe sets and contribute to the differential pattern of expression observed for the MYBL1 gene in TNBCs.

### 2.3. Analyses of the Protein Isoforms and Characterization of Sequences Demonstrating Differential Expression in MDA-M-231 Cells

Appendix A contains the complete alignment of the protein isoforms. Many of the Ref Seq isoforms demonstrate substantial sequence homologous except (a) for regions associated with their start sites, (b) the XP7 isoform, which is a truncated protein and (c) the carboxy-terminal regions of NP1, XP1, XP2 and XP3 which corresponds to the exon 15 variant (Figure 2b). The carboxy-terminal region of NP1, XP1, XP2 and XP3 (i.e., exon 15 in the variants) codes for a Negative Regulatory domain at sequences ~630–740 of the NP1 protein [17,20] (Appendix A). Modifications in this region can affect the transcriptional activity of the MYBL1 gene, thereby regulating the genes’ involvement in proliferation and differentiation signaling processes.

Part 2 of this study: Gene expression and genomic mutation analyses of MYBL1, VCPIP1, MYC and BOP1 genes localized to chromosome 8q in cell lines and patient samples

In an earlier study, we performed an unsupervised meta-analysis of GEO Affymetrix microarray datasets and identified MYBL1 as differentially expressed in a subpopulation of TNBC. As a follow-up study, we performed a knockdown of MYBL1 expression in MDA-MB-231 cells, followed by microarray analyses to determine genes that either directly or indirectly associate with MYBL1 in the TNBC [18]. Reprocessing the differentially expressed dataset, we identified genes at chromosome 8q loci that were either upregulated or downregulated following MYBL1 gene knockdown (Appendix A). Of these genes, VCPIP1, MYC and BOP1 genes were repeatedly validated as differentially expressed (with MYBL1) following transcript and protein expression analyses in MDA-MB-231 cells. The other genes did not repeatedly validate via gene expression analyses. The current study is an expansion of the previous study. In this study, MYBL1, VCPIP1, MYC and BOP1 genes are examined using additional TNBC cell lines and a patient sample dataset to validate a pattern of co-occurrence of the genes.

### 2.4. Experimental Validation of the Transcript and Protein Expression Levels of MYBL1, VCPIP1, MYC and BOP1 Genes Located at Chromosome 8q loci

Originally, we identified at least 14 genes that were both affected by knockdown of MYBL1 and located on chromosomal regions 8q12.1 to 8q23.4 loci. In the earlier experiments, all 14 genes were examined for transcript and protein expression in MCF10A (non-tumor triple-negative cell line) compared to the MDA-MB-231 TNBC cell line. Only MYBL1, VCPIP1, MYC and BOP1 were consistently validated as over-expressed in TNBC cell lines and were considered suitable candidates for further study. VCPIP1 and MYBL1 are located at the 8q13.1 locus; the MYC gene is located at 8q24.21, and the BOP1 gene at the 8q24.3 chromosomal locus. A summary of the functions of the four genes is given in Table 2. Data show that basal-like/TNBC samples display an increased percentage of gains and amplifications detected at the chromosome 8q loci [21,22,23]. The goal of the current study is to further analyze MYBL1, VCPIP1, MYC and BOP1 gene levels in an effort to determine if they might contribute to alterations identified at the 8q loci. This specific assessment will be addressed using clinical patient datasets in the section below. Meanwhile, as further validation of the reliability of the four genes, their gene expression levels were examined using additional triple-negative breast cell lines. MYBL1, VCPIP1, MYC and BOP1 transcript expression levels were assessed in MCF10A non-tumor triple negative cell lines compared to Hs 578T, BT-549 and MDA-MB-231 TNBC cell lines. PCR transcript analyses show that, consistent with previous results, TNBC cell lines express higher mRNA levels compared to the non-tumor cell line (Figure 3a). The corresponding densitometer analyses of the PCR results are shown in Figure 3b.

Protein expression analyses of MYBL1, VCPIP1, MYC and BOP1 genes were also examined in MCF10A cell lines compared to Hs 578T, BT-549 and MDA-MB-231 TNBC cell lines. The pattern of expression was somewhat consistent in that higher levels of protein were detected in TNBC compared to the non-tumor sample Figure 4a. However, Hs 549T demonstrated lower levels of VCPIP1 protein expression compared to BT-549 and MDA-MB-231, as detected via densitometer analyses (Figure 4b). This is obviously due to comparatively lower levels of protein expression, but it is unclear as to why this occurs; it might be related to the instability of the VCPIP1 protein in this TNBC cell line.

**Table 2 ijms-25-02539-t002:** Summary of the genes analyzed in this study and their biological functions.

GENE	FUNCTION
MYBL1	Proto-oncogene, a transcription factor regulating cell proliferation [24]
VCPIP1	Protein deubiquitination; Golgi mitosis
	repair; regulates protein localization to chromatin [25]
MYC	Proto-oncogene, transcription factor, cell cycle apoptosis [26,27]
BOP1	Regulation of cell cycle, p53 signal transduction class mediator [28,29]

### 2.5. Analyses of the The-Cancer-Genome-Atlas Patient Sample Dataset for Gene Expression and Genetic Alterations of MYBL1, VCPIP1, MYC and BOP1 Genes

Our previous studies focused on the validation of the genes’ expression in triple-negative cell lines. For this portion of the study, patient samples were examined as further validation of MYBL1, VCPIP1, MYC and BOP1 dysregulation in TNBCs. We searched the cBioPortal.org online resource for TNBC patient datasets in which a wide range of gene expression and genomic-mutational analyses were performed by investigators. We identified the Breast Invasive Carcinoma ‘The-Cancer-Genome-Atlas’ (TCGA) Cell 2015 dataset [30]. The dataset contains 816 patient samples, 83 of which are TNBC, as determined by immunohistochemistry. The 83 TNBC samples were examined as part of the current study, allowing for concurrent analyses of a wide range of different types of gene-related alterations. Our objective was to determine alterations in MYBL1, VCPIP1, MYC and BOP1 genes in TNBC patient samples and look for instances where one or more of the candidate genes were dysregulated with MYBL1 in particular patients. The datasets were examined using computer scripts that allow for the detection of gene expression, amplifications, homozygous and heterozygous deletions, differences in protein expression and gene fusions related to MYBL1, VCPIP1, MYC and BOP1 genes in patient samples. The precise commands utilized to identify the different alterations are given in Table 3. Under these conditions, the user can define the genes, type of alterations and cut-off values based on the experimental mean of the particular alteration. The scripts defined in Table 3 allow for the detection of the same alterations (i.e., RNA and protein analyses, amplifications, deletions and fusions) occurring in MYBL1, VCPIP1, MYC and BOP1 genes in The Breast Invasive Carcinoma TCGA 2015 dataset. Instructions on how to write the scripts are included in the user guide available at cBioPortal.org (https://docs.cbioportal.org/user-guide/oql/) Accessed on 1 February 2024.

Analyses of the Breast Invasive Carcinoma TCGA Cell 2015 dataset show significant concordance between all four genes in the 83 TNBC patients (Figure 5). MYBL1 gene is profiled in 58% of the 83 patients, the VCPIP1 gene is profiled in 57% of the patients, the MYC gene is profiled in 70% of the patients, and the BOP1 gene is profiled in 61% of the patients. Figure 5 was generated using cBioPortal software. Each rectangle represents a different TNBC patient. Alterations are defined collectively as amplifications, deletions, high mRNA or protein expression and gene fusions. Data show a somewhat similar distribution of alterations in MYBL1, VCPIP1, MYC and BOP1 in the TNBC samples. Many of the MYBL1, VCPIP1, MYC and BOP1 alterations are detected as amplifications (darker red rectangles) and higher-mRNA-expression (lighter red rectangles) in the patient samples. The light blue rectangles represent low mRNA, and the grey regions represent patients where alterations were not examined. Alterations are assumed to be absent in patients with low mRNA levels.

The data in Figure 5 were summarized and presented in Table 4. The percentage of patients profiled for alterations is presented in column 1. Column 2 shows the total number of patients in which alterations are detected. Twenty-six patients showed evidence of MYBL1 alterations, 21 with VCPIP1, 40 with MYC and 36 with alterations in the BOP1 gene. Also, in column 2 (and subsequent columns), the values in parenthesis (for VCPIP1, MYC and BOP1) represent the number of patients where both MYBL1/VCPIP1 or MYBL1/MYC or MYBL1/BOP1 alterations are detected. Compared with MYBL1, 61–84% of the same patients show alterations in VCPIP1, BOP1 and MYC genes. MYBL1 alterations are identified in 26 TNBC patients. Of the 26 patients with MYBL1 alterations, VCPIP1 alterations were detected in 61% (16/26), and MYC and BOP1 were detected in 84% (22/26) of these patients. MYC and BOP1 alterations are detected in a larger number of patients with or without MYBL1 alterations. MYBL1 and VCPIP1 followed with fewer patients demonstrating alterations; still, alterations are detected in a significant number of the same patients. These data suggest that dysregulation of MYC exerts a stronger signaling effect compared to the MYBL1 gene. Nonetheless, there is an obvious relationship between all four genes located at chromosomal regions 8q13.1–8q24.3 in the TNBC patient samples.

The number of patients with amplifications in MYBL1, VCPIP1, MYC and BOP1 are shown in column 3. For some genes, amplifications are detected in the same patients. Ten patients show MYBL1 amplifications and VCPIP1 amplifications are detected in the same 10 patients (Table 4). MYC amplifications were detected in 8/10 of these patients, and BOP1 amplifications in 7/10 of these patients.

The number of patients with high mRNA levels for our genes is shown in column 4, and low mRNA levels are in column 5. Higher numbers of patients display high mRNA levels of MYBL1 and BOP1 gene, then VCPIP compared to MYC. A limited number of missense or deletion mutations were detected. But overall MYC and BOP1 alterations were detected in a larger number of patients compared to MYBL1 and VCPIP1 genes. Collectively, these data demonstrate a significant number of different types of alterations in MYBL1, VCPIP1, MYC and BOP1 genes in some of the same patients. The role of the MYC gene in TNBC is well documented. But, the combined involvement of MYBL1, VCPIP1 and BOP1 in the same patient samples has not been documented. All four genes are located at the chromosome 8q loci, and likely all contribute to the high-level alterations found in this region. This is a small study, so we cannot draw definitive conclusions, but the results of the study support further validation using larger datasets of molecularly characterized TNBC patient samples.

The MYC gene is likely key to processes involving MYBL1, VCPIP1 and BOP1. However, this does not discount the possible contribution of the MYBL1 gene. Our knockdown study shows that when MYBL1 is knocked down, MYC expression is downregulated, supporting a previous observation of MYBL1 regulation of MYC expression [1]. It could be that when the MYBL1 gene is knocked down, MYC is downregulated, thereby affecting BOP1 and maybe even VCPIP1 expression. We identified VCPIP1 expression as dysregulated in TNBCs in our studies and considered a possible relationship with the MYBL1 gene. Along with the gene’s involvement in mitosis, the VCPIP1 gene is a deubiquitinating protein (DUB) that plays a role in protein turnover. The DUB family of genes is being considered as therapeutic targets for cancer [31]; hence, VCPIP1 is an interesting candidate to examine for its possible involvement in TNBC. This study is limited to analyses of TNBC, but evaluations of our gene panel in ProteinAtlas^TM^ show a distinct pattern of expression between MYBL1 and VCPIP1. Figure 6 is a TCGA Pan-Cancer RNA-sequencing profile extracted from ProteinAtlas^TM^ [32,33]. The data summarize analyses of RNAs in 17 different types of cancers. The RNA Seq values are low but still appear to show a similar pattern of expression in MYBL1 and VCPIP across cancers, with both genes demonstrating peak expression in breast samples (without distinctions between breast cancer subtypes). MYBL1 and VCPIP1 involvement in cancer has been detected in another study. Bubola et al. [34] detected MYBL1/VCPIP1 fusion products in adenoid cystic carcinoma of the salivary glands. The current study is the first suggestion of a possible relationship between MYBL1 and VCPIP1 in TNBC. Analysis of MYC and BOP1 gene profiles in the ProteinAtlas^TM^ 17 pan-cancer dataset displays a similar pattern of RNA-seq levels in breast cancers.

The STRING^TM^ program can be used to graphically display protein: protein interactions via linkages based on documented citations and predictions. MYBL1, VCPIP1, MYC and BOP1 were processed using STRING^TM^ in an effort to demonstrate known and predicted interactions between the genes (Figure 7). Although the current focus is to compare the alterations based on comparison to MYBL1, based on published and predicted data, the strongest connections between MYBL1, VCPIP1 and BOP1 genes appear to be via the MYC gene as demonstrated by its central location in the linkage display. There are substantial data showing direct biological connections between MYBL1 and MYC [13], and MYC and BOP1 [35], and an indirect association between MYC and VCPIP1 as predicted by STRING^TM^. The relationship between MYBL1 and MYC is based on co-expression and text mining. The relationship between MYC and BOP1 is based on their close proximity, experimental validation and co-expression. MYC gene is located at chromosomal location 8q24.21, and the BOP1 gene is located at 8q24.3. There is an indirect relationship between MYC and the VCPIP1 gene via the Valosin-containing protein (VCP) [36]. The VCP gene is involved in DNA repair, replication and regulation of the cell cycle, similar to MYC and MYBL1 genes. Data suggest that MYC is a putative VCP substrate. We propose that VCPIP1 and MYC are linked via the NSFL1C gene also present on the STRING^TM^. As part of a preliminary study of VCPIP1 interacting genes, we find NSFL1C substantially differentially expressed (unpublished experimental observation) in TNBC. MYBL1, VCPIP1, MYC and BOP1 are the only genes in the STRING^TM^ on chromosome 8q.

Related to pathways and processes, there is only one known direct linkage between MYBL1, VCPIP1, MYC and BOP1. MYBL1 and MYC genes are linked via the pathway associated with adenoid cystic carcinoma. Individually, the BOP1 is regulated by signal transduction by p53 mediator, and VCPIP1 is associated with Deubiquitinate pathways. Considering all of the genes displayed in the STRING^TM^, a large number are related by ‘Ubi Conjugation’. These proteins are post-translationally modified by ubiquitin, SUMO, APG12, URM1, or RUB1 [37].

MYBL1 and VCPIP1 are both located at the chromosomal 8q13.1 locus. Bubola et al. observed fusion products between MYBL1 and VCPIP1 [34] in salivary gland tumors; however, no fusion alterations were detected in this smaller TNBC dataset. Data from our studies show VCPIP1 affected by the knockdown of MYBL1 in MDA-MB-231, concordant expression of both genes in a particular patient sample dataset and increased gene expression in some TNBCs. Still, it could be that the downregulation of VCPIP1 (following the knockdown of MYBL1) is due to a more direct regulation by MYC. We are exploring the possibility that MYBL1 and/or MYC transcription factors bind to the VCPIP1 promoter and directly regulate the gene.

**Figure 6 ijms-25-02539-f006:**
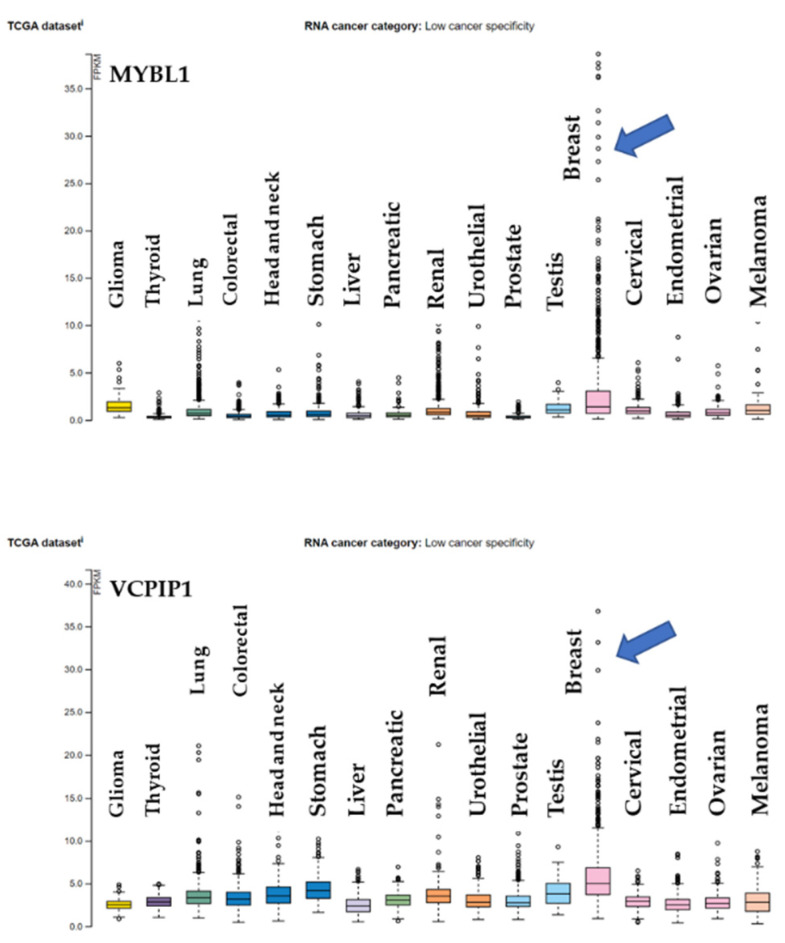
A pan-cancer study using RNA Sequencing data extracted from ProteinAtlas.org Accessed 1 February 2024. MYBL1 and VCPIP1 RNA-seq expression data were generated from Pan-Cancer TCGA patient samples in various types of cancers [32]. The MYBL1 (top panel) and VCPIP1 (bottom panel) RNA Seq levels for breast cancers are noted by the arrow. MYBL1 and VCPIP1 show a similar pattern of expression in analyses of breast cancers compared to other cancer types. Data suggest the genes demonstrate low overall cancer specificity as pan markers. FPKM is Fragments Per Kilobase of transcript per Million mapped reads.

**Figure 7 ijms-25-02539-f007:**
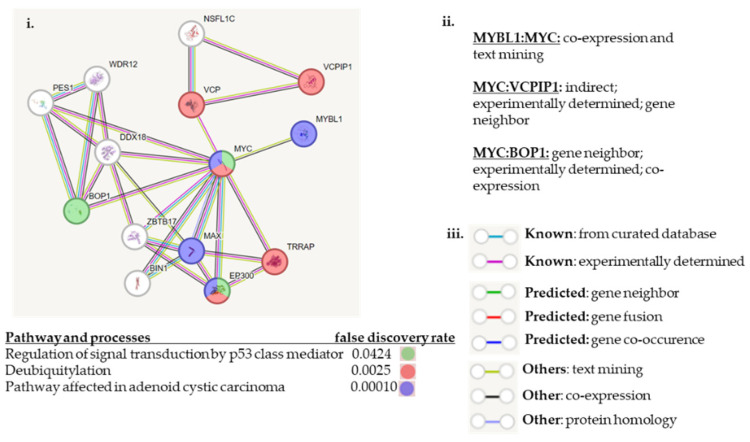
STRING^TM^ analyses of MYBL1, VCPIP1, MYC and BOP1 protein: protein associations. These data display linkage analyses of predicted and experimental protein: protein interactions between MYBL1, VCPIP1, MYC and BOP1 genes [38]. (**i**) the linkage analyses between MYBL1, VCPIP1, MYC and BOP1 genes are shown here. (**i**,**ii**) show that there is a direct relationship between MYBL1 and MYC genes based on co-expression and text mining. And there is a direct relationship between MYC and BOP1 based on gene location, experimental validation and co-expression. And there is an indirect relationship between MYC and VCPIP1 via the Valosin-containing protein (VCP) gene. (**iii**) Summarizes the legend for the colored linkages.

## 3. Discussion

The primary focus of the current study is to further characterize the MYBL1 gene in TNBC. Although treatments for these and other cancers have improved over the past few years, with the emergence of antibody-drug conjugate and immunotherapeutic approaches [39], it is still important to characterize the cancers on a molecular level. TNBC is characterized based on three molecular signatures, but the cancers are extremely heterogeneous and can be further divided into at least six subcategories [40]. Because the cancers often express immune signatures, we performed unsupervised meta-analyses of GEO TNBC microarray datasets and then further subdivided the cancers based on immune signatures, which subsequently led to the identification of MYBL1 differential expression. Data showed that an estimated 25% of TNBC demonstrated differential expression of MYBL1. Because MYBL1 is a strong transcription factor known to be involved in (and likely affect) events related to the pathogenesis of cancer, we are devoting studies to better understanding the gene in TNBC.

In the current study, our first approach was to examine transcript variants and isoforms that define the human MYBL1 gene deposited at the NCBI. Data presented in this study represent the first analyses of the human MYBL1 Ref Seqs. The results of this study will help in future experiments in the event there is a need to identify functional variants and isoforms of the gene in a particular disease. The alignment of the sequences shows substantial similarity. There are nine unique MYBL1 Ref Seqs (NM1, NM2/3, XM1, XM2, XM3, XP4, XM5, XM6 and XM7). The NMs represent the curated Ref Seqs, and the XMs represent predicted contig Ref sequences. The most substantial differences between the Ref Seqs are in the 5′ region of the gene and near the end of the coding region of the gene, where there is a unique exon 15 in NM1 and 3 of the predicted sequences. The most dramatic differences are between NM1, NM2 and XM7 sequences. In MYBL1 proteins, sequences corresponding to exon 15 are in the carboxy-terminal region, which includes sites allowing for phosphorylation, ubiquitination and acetylation post-translational events involved in transactivation events [20]. Aside from structural alterations that lead to changes in the gene’s performance, transient changes in events associated with the carboxyl terminus might contribute to effects on MYBL1 gene expression in cancers. The relevance of this region in TNBC can best be deciphered via studies that focus on targeted analyses of the region. As for the predicted MYBL1 sequences, the XM1/XP1, XM2/XP2 and XM3/XP3 align closely with NM1/NP1 at exon 15 but differ at their start sites. XM7/XP7 is a truncated version of the other sequences, and if it is determined to be functional, we suspect it will have a different function in TNBC.

MYBL1 is a strong transcriptional regulator. Following MYBL1 knockdown, we identified an enrichment of genes associated with MYBL1 transcriptional regulation [18], and a number of differentially expressed genes located at chromosome 8q loci affected either directly or indirectly by the knockdown process. Other studies report frequent copy number gains, high-level amplifications and mutations at the 8q loci [21,41]. Genes and siRNAs key to cell growth, survival and transformation have been identified at these loci [42] as well. The 8q loci were of particular interest to us because of observations made in preliminary bioinformatic-based studies. MYBL1 gene is located at chromosome 8q13.1, and bioinformatic analyses of different TNBC datasets showed significant Spearman correlations between the expression of MYBL1 and VCPIP1, which is also located at chromosome 8q13.1. Subsequent supervised analyses of our and other datasets led to the identification and selection of VCPIP1, MYC and BOP1 associated with MYBL1 as candidate genes to study in TNBCs. The role of the MYC gene at regions near 8q12–8q24 is well established [22,43,44]. Experimental validations of transcript and protein expression analyses seemed to substantiate this relationship. For the current studies, it was important for us to examine our candidate genes for the concurrent analyses of a wide range of alterations, recognizing that dysregulation comes in many forms. The Breast Invasive Carcinoma TGCA 2015 study allowed for this approach. Examining patients in this dataset, we found that different types of alterations in MYBL1, VCPIP1, MYC and BOP1gene occur in a substantial number of the same TNBC patients. At this point, it is unclear if there are direct signaling events linking all four genes, but these data suggest there might be. The STRING^TM^ linkage analyses show direct connections via MYC to MYBL1 and BOP1 genes and indirect associations with the VCPIP1 gene. We are exploring the possibility that the MYBL1 gene can also drive these processes.

Analyses of clinical patient sample datasets appear to support a relationship between MYBL1 and VCPIP1 genes. Utilizing the TGCA 2015 TNBC dataset, we find significant concordance between patients with MYBL1 and VCPIP1 alterations, including amplifications and high mRNA expression. Both genes are located at the 8q13.1 locus. Even though this is a small sample size of 83 TNBC, the numbers are significant enough to make a case for further analysis. The MYBL1 gene is associated with cell cycle and proliferation events [45], and VCPIP1 is involved in DNA repair, deubiquitinating events related to the reassembly of Golgi apparatuses and endoplasmic reticulum following mitosis [46]. It could be that there are signaling events linking the MYBL1 (cell proliferation) and VCPIP1 (mitotic signaling) genes. Even though there are a larger number of patients with MYBL1, MYC and BOP1 alterations compared to VCPIP1, data suggest a relationship with the VCPIP1 gene. The NSFL1C and VCPIP1 genes are involved in the regulation of the Golgi membrane dynamics associated with cell division [47]. STING^TM^ showed a connection between both genes. And, preliminary experiments in our laboratory show reliable differential expression of NSFL1C and VCPIP1 genes in TNBC (unpublished experimental observations). STRING^TM^ suggests this relationship is more closely linked to MYC. It could be, but we are examining the possibility that the MYBL1 transcription factor directly binds to VCPIP1 promoter allowing for direct regulation in TNBC. MYBL1 potential transcription factor binding sites are identified in the promoter region of the VCPIP1 gene, and bioinformatic analyses show there is a strong gene-association-score connecting both enhancers and MYBL1 to the promoter region of VCPIP1 gene (GeneCards [48]/‘Gene-targets-for-GeneHancer’ location (https://www.genecards.org/cgi-bin/carddisp.pl?gene=VCPIP1&keywords=vcpip1, accessed on 1 February 2024). Related to other cancers, MYBL1/VCPIP1 gene fusions were detected in salivary gland tumors [34] supporting a documented relationship between the two genes.

There is a substantial amount of research documenting the role of the MYC gene in TNBC [49,50,51,52,53,54]. The MYC transcription factor is located at the far end of chromosome 8q at the 8q24.21 locus, and the MYBL1 transcription factor is located at the 8q13.1 locus. Both MYBL1 and MYC are involved in cell growth and proliferation signaling, supporting the cooperation of the two genes in these processes. Data show that when MYBL1 is knocked down in MDA-MB-231 cells, MYC is downregulated. The results of these data are consistent with investigators who find that the MYBL1 transcription factor regulates MYC [1]. All TNBC cell lines did not necessarily show the same pattern in protein expression. We are studying a larger panel of TNBC, and preliminary analyses show that some TNBC cells express MYBL1 and MYC, and some do not. These data emphasize the heterogeneity of TNBC. Previous studies document co-expression of MYBL1 and MYC genes. Co-expression of MYBL1 and MYC are detected in pharynx cancers [55], adenoid cystic carcinoma [56] and breast, ovarian and lung cancers [13,45,57,58]. For the most part, data presented in this study are consistent with previous observations showing over-expression and amplification of MYC in breast cancers [23,59,60], with a notable correlation between MYBL1 and MYC amplification alterations in the TCGA 1015 patient sample TNBC dataset.

The BOP1 gene is located at the 8q24.3 locus nearer the MYC gene locus. The BOP1 gene facilitates RNA binding processes related to the regulation of the cell cycle [61]. Data show that the gene is over-expressed in tumor progressions [62], confers chemoresistance in TNBC samples [63], and functions via mitogen-activated protein kinase (MAPK) signaling [64]. In the current study, the BOP1 gene was over-expressed in TNBC cell lines [18] and patient samples along with MYBL1, VCPIP1 and MYC genes. Patient sample data show a close relationship between MYBL1 and BOP1; however, in the GENE description of BOP1 located at NCBI, 175 genes are defined as interacting with BOP1, one of which is MYC gene based on affinity capture Mass Spectrometry (https://www.ncbi.nlm.nih.gov/gene/23246, accessed on 1 February 2024); there is no mention of MYBL1. The association might be of a different sort. Nonetheless, investigators detect MYC and BOP1 co-expression in hepatocellular carcinoma [35], and we observe MYC and BOP1 co-expression in some TNBC. BOP1 and MYC alterations are detected in a substantial number of patients. In TNBC, it could be that MYBL1 regulates MYC, with MYC then regulating BOP1. These interactions can account for higher numbers of patients with MYC and BOP1 alterations compared to patients with MYBL1 and VCCPIP1 alterations. The BOP1 gene is located near the terminal region of chromosome 8q within proximity to the MYC gene. It could be that this is a hotspot for alterations, more so than internal 8q loci. Data obtained using STRING^TM^ analyses show documented relationships between MYBL1 and MYC and MYC and BOP1.

## 4. Materials and Methods

### 4.1. Cell Lines and Patient Sample Datasets

*Cell lines*: The cell lines utilized in this study were purchased from the American Type Culture Collection (ATCC; Old Town Manassas, VA, USA) ATCC.org and utilized within 6 months of purchase. The cell lines include MCF10A, which is a triple-negative non-tumor cell line, and Hs 578T, BT-549 and MDA-MB-231, all of which are mesenchymal-like TNBC cell lines. The cells were fed twice weekly with Dulbecco’s Modified Eagle Minimum essential media (DMEM, Thermo-Fisher Scientific, Waltham, MA, USA) supplemented with 1% penicillin-streptomycin solution and 10% fetal calf serum and grown in a 37 °C incubator (NuAire Laboratory Equipment, Plymouth, MN, USA) with additional 5% CO_2_. The penicillin–streptomycin and fetal calf serum were purchased from the ATCC. The cells were grown until 80–90% confluent and trypsinized using a 0.25% Trypsin-EDTA solution (Gibco^TM^, Thermo Fisher Scientific, Waltham, MA, USA), at which time they were suitable for sub-culturing or experimental studies.

*Patient sample datasets*: Data corresponding to TNBC patient samples were obtained from the cBioPortal.org [65] website. The website is an interactive open-source website that contains thousands of curated cancer patient samples collected from different databases, developed and supported by Memorial Sloan Kettering Cancer Center (MSK), Dana Farber Cancer Center, Princess Margaret Cancer (Toronto, ON, Canada), and Children’s Hospital of Philadelphia and other academic and clinical facilities. The cBioPortal database contains high-quality molecular profiles with substantial amounts of supporting clinical attributes. For the current study, the database was searched based on the type of cancers and preference was given to datasets where different types of alterations were examined in patient samples. Alterations are defined collectively as amplifications, deletions, high mRNA or protein expression and gene fusions. The Breast Invasive Carcinoma The-Cancer-Genome-Atlas (TCGA) 2015 patient dataset was selected. The Breast Invasive Carcinoma TCGA 2015 dataset contains 818 breast cancer samples [30]. Eighty-three TNBC samples (previously characterized based on immunohistochemistry) were extracted from the patient dataset and utilized for the current study; TNBCs were not divided into subcategories. The Breast Invasive Carcinoma TCGA 2015 dataset was chosen because patient samples were examined for a wide range of alterations, including RNA and protein expression levels and various types of genomic analyses, including homozygous and heterozygous deletion, gene fusion and amplification analyses. This dataset allows for the simultaneous analyses of a wide range of alterations associated with candidate genes.

### 4.2. Ribonucleic Acid (RNA) Isolation, cDNA Generation, PCR and Gene Analyses

Cell lines utilized for this study were grown to approximately 90% confluency in T75 dishes, and total RNA was extracted using the Trizol^TM^ solution (Thermo Fisher Scientific, Waltham, MA, USA) as suggested by the manufacturer. Samples displaying intact 28S/18S RNA profiles (as determined by gel electrophoresis) and A260/280 spectrophotometer absorbances with ratios of 1.8–2.0 are used to generate cDNA preparations. The SuperScript III^TM^ reverse transcriptase (Invitrogen, Waltham, MA, USA) kit containing oligo (dT) primers was used to generate first-strand cDNA. The procedure was performed as recommended by the manufacturer. Conventional polymerase chain reaction (PCR) was performed in order to generate gene-specific amplicons as a measure of mRNA transcript levels. The PCR reactions each contained 2 uL (~0.5 μM) of forward and reverse primers, 2 μL of cell-line-specific cDNA, 10 μL of 2X thermostable DNA polymerase I TAQ polymerase master mix (optimized for TAQ enzyme, TAQ buffer and dNTPs; Life Technologies, Carlsbad, CA, USA), and water up to 20 μL. The PCR reaction was placed in PCR-quality tubes and processed using the Bio-Rad Thermal Cycler (Bio-Rad Laboratories, Hercules, CA, USA). The cycler conditions were (a) 5 min at 95 °C degrees, (b) 30 cycles for 30 s at 95 °C, followed by 30 s at 58 °C degrees, then 30 s at 72 °C. The ensuing amplicons were separated by gel electrophoresis, and their signal intensities were quantitated and analyzed using the Li-COR Image system software (Li-COR Biosciences, Lincoln, NE, USA).

### 4.3. PCR Gene Primer Sets Generation

The Primer3^TM^ [66] program was used to generate primers for each of the target genes. The primers were designed based on the probe-set sequences on the U133 plus 2 Affymetrix microarray platform. The nucleotide sequences corresponding to genes on the microarray are available as downloads from ThermoFisher.com (USA). The primer sequences for the control and 4 final candidate genes, GAPDH (control), BOP1, VCPIP1, MYC and MYBL1, are given below (Table 5). The nucleotide sequence and size of the resulting amplicons were examined and validated using the In-Silico *PCR^TM^* online tool available on the University of California Santa Cruz web browser [67]. The primer sets were used for comparative analyses of gene expression between MCF10A, Hs 578T, BT-549 and MDA-MB-231 cell lines.

### 4.4. Data Analyses, Including Sequence Alignment Comparison and the cBioPortal Open Source Database

*Sequence alignment*: The MultAlin^TM^ program [68] was used to perform sequence alignment analyses between the MYBL1 transcript variants and protein isoforms retrieved from the National Center for Biotechnology Information (NCBI) [16]. The sequences were determined by Gene Assembly GRCH38.p14. Individual exons associated with the transcript variants were compared to Ref Seq NM_001080416.4 (which was the longest variant).

*Patient sample analysis using cBioPortal*: We searched cBioPortal.org [65] for TNBC patient sample datasets containing datasets previously examined for a wide range of genomic, transcriptome and proteomic alterations. The Breast Invasive Carcinoma TCGA 2015 study [30] was identified. Patient samples in the TCGA 2015 study were previously examined via whole exome DNA sequencing, RNA sequencing, and gene analysis platforms, allowing for analyses of different types of genomic mutations and gene expression comparisons. The cBioPortal.org online resource was used to perform comparative analyses of our signature genes in TNBC patient samples. Directions for writing the computer scripts (i.e., Onco Query Language) allowing for identification of particular types of genomic anomalies associated with the patient samples are defined in the user guide available at cBioPortal.org (https://docs.cbioportal.org/user-guide/oql/, accessed on 1 February 2024). The precise scripts used for these analyses are given in the form of a table in the Section 2. All 4 genes are analyzed for the same type of alterations in the patient datasets; hence, the scripts are identical.

### 4.5. Data Analyses of Protein-Protein Interaction Using STRING^TM^ Open Source Database and PhosphoSitePlus^TM^

The Search Tool for the Retrieval of Interacting Genes/Proteins (STRING^TM^) was utilized to determine known and predicted interactions between genes validated as part of this study [38]. The STRING^TM^ tool utilizes millions of data points validated by experimental, text mining and bioinformatic analyses to predict protein-protein interactions of candidate genes. PhosphoSitePlus^TM^ was utilized to reveal the location of the Regulatory domain in the carboxy-termini of MYBL1 protein [19].

### 4.6. Western Blotting

The western blot procedure was performed as outlined in a previous document [12]. The estimated molecular weight of the proteins was determined using the Novex™ Sharp Pre-stained Protein Standard (LC5800; Thermo Fisher Scientific, Waltham, MA, USA). Antibodies used in the current study are outlined below: *Antibodies*: MYBL1 anti-mouse monoclonal antibody (clone 2A2) was utilized at a dilution 1:500 (SAB14002280; Sigma Aldrich/Millipore, St. Louis, MO, USA); sequence listed below: _KSLVLDNWEKEESGTQLLTEDISDMQSENRFTTSLLMIPLLEIHDNRCNLIPEKQDINSTNKTYTLTKKKPNPNTSKVVKLEKNLQSNCEWETVVYGKTEDQLIMTEQAR_. This sequence aligns with the carboxy-termini of NP1 protein and exon 15 corresponding region. Actin anti-rabbit polyclonal antibody was used at a 1:1000 dilution (SAB14002280; Sigma Aldrich/Millipore, St. Louis, MO, USA). MYC anti-mouse polyclonal antibody (SC-40), BOP1 anti-mouse monoclonal antibody (Sc-365595) and VCPIP1 anti-mouse monoclonal antibody (sc-515291) were purchased from Santa Cruz Biotechnology (Santa Cruz, CA, USA) and used at a 1:500 dilution. Secondary Mouse IgG Horseradish Peroxidase-conjugated (HRP) conjugated anti-mouse antibody (HAF007) and Secondary Rabbit IgG Horseradish Peroxidase-conjugated (HRP) conjugated anti-rabbit (HAF008) antibodies were purchased from R and D Systems (Minneapolis, MN, USA) and utilized at a 1:1000 dilution. Western blotting filters were processed and developed using the Clarity Western ECL substrate (Bio-Rad Laboratories, Hercules, CA, USA) on the LI-COR digital imaging system (LI-COR Biosciences, Lincoln, NE, USA). Molecular weight estimation.

## 5. Conclusions

To conclude, in the current study, (a) we characterize both the curated and predicted variants and isoforms described for the MYBL1 gene and (b) identify VCPIP1, MYC and BOP1 genes (all of which are located at 8q loci) as reproducibly dysregulated with MYBL1 in TNBC cell lines and patient samples. We suggest that collectively, these genes contribute to the high incidence of alterations influencing the TNBC genotype. We are aware that TNBCs represent a comparatively smaller percentage of breast cancer patients, and our data suggest that alterations are detected in yet a fraction of these patients. All the same, we find that the genes are altered and likely contribute to processes in TNBCs. And based on the reproducibility of these data, we strongly suspect that together, these genes are connected via cell signaling events affecting TNBCs.

## Figures and Tables

**Figure 1 ijms-25-02539-f001:**
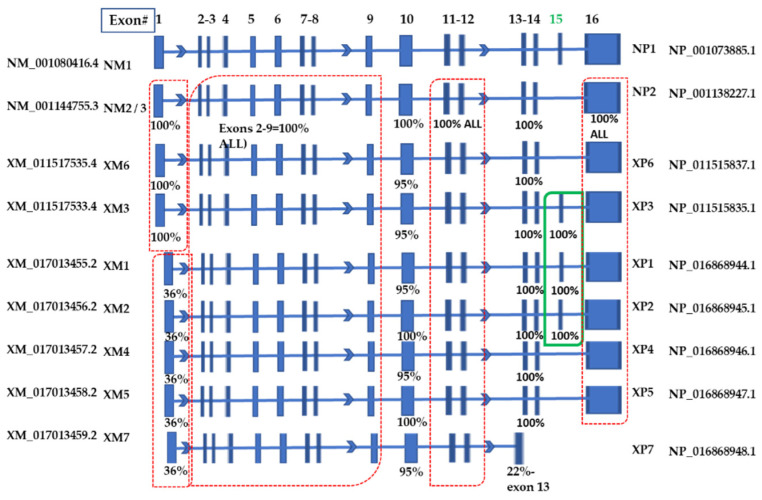
MYBL1 transcript variant alignments. Both curated and predicted sequences associated with the MYBL1 gene are included. The NM1 Ref Seq is the longest transcript variant, and the exon sequences associated with each are compared to the NM1 exon nucleotide sequences. The position of the unique exon 15 is in the green box. The corresponding protein isoforms are noted to the far right of the figure.

**Figure 2 ijms-25-02539-f002:**
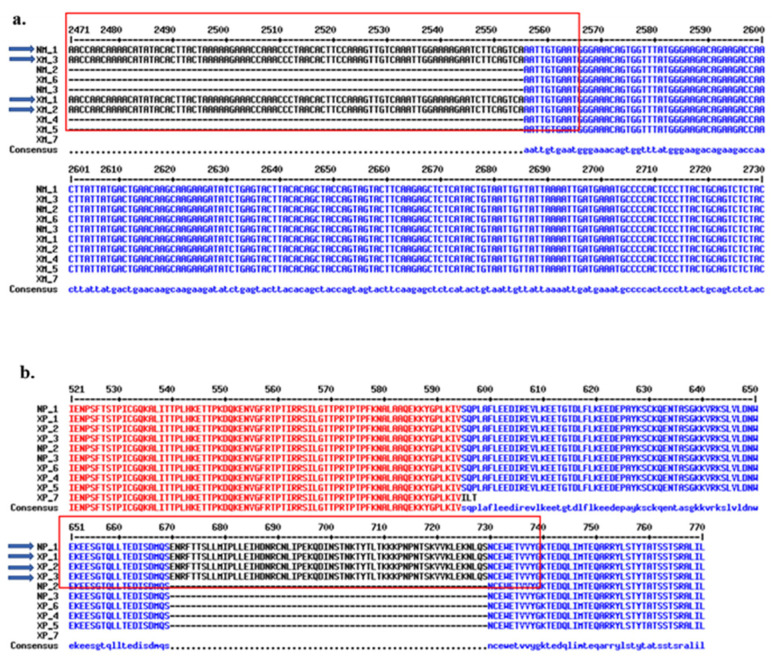
Nucleotide Variant and Protein isoform sequence alignment demonstrating the location of exon 15 sequences in the variants and corresponding isoforms. (**a**) The regions related to exon 15 for the nucleotide variants (NM1, XP1, XP2, XP3) are outlined in the red box, and (**b**) the protein isoforms (NP1, XP1, XP2 and XP3) corresponding to the exon 15 region are in the red box. The arrows designate the variants that contain exon 15 or isoforms that contain the protein region corresponding to exon 15. Black sequences represent partial sequence similarity (i.e., 4 of 10 Ref Seqs), and blue sequences represent ‘nearly all similar sequences’ (i.e., 9 of 10 similar). Red sequences represent 100% similarity between all queries. The numbers in the heading represent the number of (**a**) nucleotides or (**b**) amino acids from the start site of NM1 and NP1 Ref Seq, respectively. Alignments of all 10 transcript (**a**) variants and (**b**) protein isoforms were generated using the MultAlin^TM^ program. Note that some of the sequences do not align with the sequences contained within the red boxes (as designated by dashed lines representing missing sequences).

**Figure 3 ijms-25-02539-f003:**
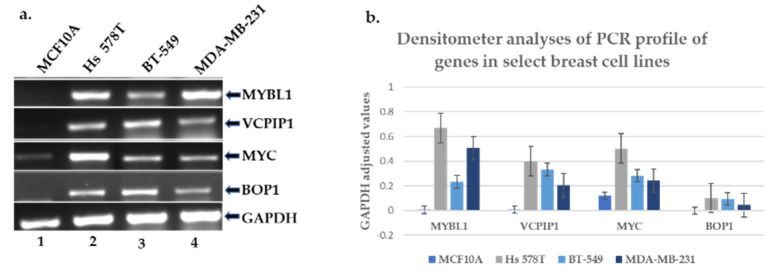
PCR transcript analyses of chromosome 8q candidate genes in breast triple-negative cell lines. (**a**) PCR followed by gel electrophoresis was performed to determine the transcript levels of MYBL1, VCPP1, MYC and BOP1 genes in the MCF10A non-tumor triple-negative cell line (lane #1) compared to Hs 578T (lane #2), BT-549 (lane #3) and MDA-MB-231 (lane #4) TNBC cell lines. (**b**) represents the densitometer analyses of the corresponding PCR profiles. Probe sets were designed based on regions defined by the Affymetrix microarray platforms.

**Figure 4 ijms-25-02539-f004:**
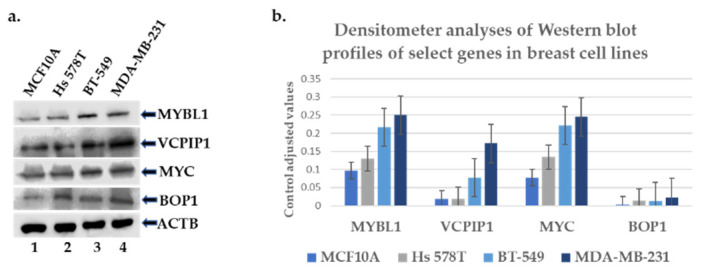
Western blot protein analyses of chromosome 8q candidate genes in breast cell lines. (**a**) Western blot experiments were performed to determine the protein levels of MYBL1, VCPP1, MYC and BOP1 genes in the MCF10A non-tumor triple-negative cell line (lane #1) compared to Hs 578T (lane #2), BT-549 (lane #3) and MDA-MB-231 (lane #4) TNBC cell lines. (**b**) Represents the densitometer analyses of the corresponding western blot profiles.

**Figure 5 ijms-25-02539-f005:**
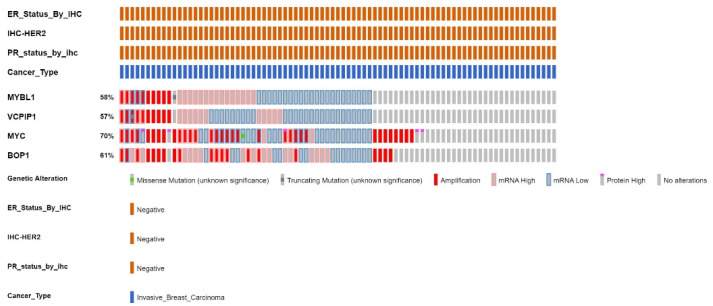
cBioPortal analyses of The Breast Carcinoma TCGA 2015 TNBC patient sample datasets for alterations in MYBL1, VCPIP1, MYC and BOP1 gene. Each rectangular block corresponds to a different TNBC patient (i.e., 83 total TNBC patients in the dataset). The ‘percent values’ correspond to the percentage of patients in the dataset that were profiled or analyzed. Alterations were not detected in all patient samples, and all patient samples were not examined. The legend for the alterations is listed below the patient sample profile. All samples are breast invasive carcinoma TNBC. Profiles and values were generated using cBioPortal (https://www.cbioportal.org/).

**Table 1 ijms-25-02539-t001:** Summary of MYBL1 Ref Seq transcripts and isoforms. The transcripts and isoforms corresponding to the human MYBL1 gene are listed below. The official names designated by NCBI are abbreviated. The sizes and start sites for each nucleotide (variant) and protein (isoform) are based on data available from the NCBI.

NUCLEOTIDE	Abbreviation	SIZE (Base Pairs)	Start Site (bp)	PROTEIN	Abbreviation	SIZE (Amino Acids)
**NM_001294282.1**	**NM3**	4878	381	**NP_001281211.1**	**NP3**	691
**NM_001144755.3**	**NM2**	4981	381	**NP_001138227.1**	**NP2**	692
**NM_001080416.4**	**NM1**	5161	381	**NP_001073885.1**	**NP1**	752
**XM_011517535.4**	**XM6**	4996	381	**NP_011515837.1**	**XP6**	697
**XM_011517533.4**	**XM3**	5176	381	**NP_011515835.1**	**XP3**	757
**XM_017013455.2**	**XM1**	5054	220	**NP_016868944.1**	**XP1**	770
**XM_017013456.2**	**XM2**	5036	217	**NP_016868945.1**	**XP2**	765
**XM_017013457.2**	**XM4**	4871	217	**NP_016868946.1**	**XP4**	710
**XM_017013458.2**	**XM5**	4853	241	**NP_016868947.1**	**XP5**	705
**XM_017013459.2**	**XM7**	2270	222	**NP_016868948.1**	**XP7**	597

**Table 3 ijms-25-02539-t003:** Scripts used for detection of gene alterations related to Breast Invasive Carcinoma TCGA 2015 dataset: The same scripts allow for detection of the same alterations (i.e., RNA and protein analyses, amplifications, deletions and fusions) occurring in MYBL1, VCPIP1, MYC and BOP1 genes. For each alteration, experimental values 2-fold above and below the mean were tallied.

**MYBL1**: PROT>=2 PROT<=-2 AMP HOMDEL EXP>2 EXP<2 MUT FUSION HETLOSS;
**VCPIP1**: PROT>=2 PROT<=-2 AMP HOMDEL EXP>2 EXP<2 MUT FUSION HETLOSS;
**MYC**: PROT>=2 PROT<=-2 AMP HOMDEL EXP>2 EXP<2 MUT FUSION HETLOSS;
**BOP1**: PROT>=2 PROT<=-2 AMP HOMDEL EXP>2 EXP<2 MUT FUSION HETLOSS;

**Table 4 ijms-25-02539-t004:** Summary of the alterations in MYBL1, VCPIP1, MYC and BOP1 genes generated from analyses of cBioPortal patient sample datasets in Figure 5. The Percent Alterations represent the percentage of patients profiled. The total alterations, amplifications, high/low mRNA or no-alterations are also included. The values in parenthesis (for VCPIP1, MYC and BOP1) represent the number of patients where both MYBL1/VCPIP1 or MYBL1/MYC or MYBL1/BOP1 alterations are detected.

	%ALTERATIONS /Profiled	# Patients/ALTERATIONS	AMPLIFICATIONS	HIGH mRNA	LOW mRNA	NONE
**MYBL1**	58% of 83	26 patients	10/83	17/83	25/83	36/83
**VCPIP1**	57% of 83	21 patients (16 same)	9/83 (9 same)	14/83 (8 same)	28/83 (18 same)	37/83 (36 same)
**MYC**	70% of 83	40 patients (22 same)	32/83 (8 same)	11/83 (7 same)	28 (15 same)	25/83 (25 same)
**BOP1**	61% of 83	36 patients (22 same)	20/83 (7 same)	26/83 (14 same)	16/83 (12 same)	32/83 (31 same)

**Table 5 ijms-25-02539-t005:** PCR primer sequences.

A: GENE	LEFT PRIMER	RIGHT PRIMER	Size of Amplicon (Bp = Base Pair)
**MYBL1**	TGGATAAGTCTGGGCTTATTGG	CCATGCAAGTATGGCTGCTA	210 BP
**MYC**	TGGCTGCTTGTGAGTACAGG	TGAACTGGCTTCTTCCCAGG	229 BP
**BOP1**	CACCCCCAGCTTCTATGACC	CTGGATGAAGCGTCCGTAGG	256 BP
**VCPIP1**	AGGACATTAAGCGGGCCAAT	GGGAACACCCTCAGGTGGTA	266 BP
**GAPDH**	TCCCTGAGCTGAACGGGAAG	GGAGGAGTGGGTGTCGCTGT	217 BP

## Data Availability

The data is available from the PI (A.P.) upon request.

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
