# Peer review of "Characterization of MYBL1 Gene in Triple-Negative Breast Cancers and the Genes’ Relationship to Alterations Identified at the Chromosome 8q Loci"

_ijms, 2024, doi:10.3390/ijms25052539_

Round 1

Reviewer 1 Report

Comments and Suggestions for Authors

Dear Authors,

minor but important changes will improved this interesting manuscript:

1. please check carefully and correct font's colour, becouse in some part of manuscript is grey (foe examlpe L38-39 and onther) or fragments have been pasted (section keywords).

2. in materials and methods, it should be put in order information about company, city and also country of reagents; information of country is often missing (name of state in USA is non satisfactory for Reviewer from Europe, trust me). I section cell line there is no information about reagents/supplement's company, please add tthis information.

3. Correct notation of one of the cell lin is MDA-MB-231, according to ATCC (ATCC, American Type Culture Collection, Old Town Manassas, VA, USA).  Hovewer on Figures shorter version is acceptable, please correct this in manuscript and also in case of BT-549 cell line.

Reviewer 2 Report

Comments and Suggestions for Authors

Dear Authors,

The article Characterization of MYBL1 gene in triple negative breast cancers and the genes’ relationship to alterations identified at the 3 chromosome 8q loci” introduces characterization of the MYBL1 and its possible role in TNBC pathogenesis and its connection with the genes VCPIP1, MYC and BOP1.

   The article is not well-structured (it is not obvious the division of the article on 2 parts): better to follow standards of the manuscripts. It would be better to introduce computer analysis data of Part 1 as the chapter of the whole article.

Tables and figures are not carefully illustrated, the scale should be increased. Photos and figures in the Supplementary section have a good quality.

 Material and methods section contain the chapter “Cell lines and Patient samples”. But there were no any real patient samples studied; they have been taken from the Cancer Genome Atlas. It should be mentioned in the text in a clearer manner. The real experiments were only PCR and Western blot on cell lines.

The English is not well-understandable: present indefinite is used instead of past indefinite or past perfect: “we analyze”, “we find” etc.

1.       Table 1 in Part 1 is absent. There is table 1 in the MATERIALS AND METHODS section and contains primers.

2.       Table 3 does not contain any references

3.       Table 4 has no sense, all the columns are identical

4.       Table 5 is a screenshot, the quality of the table is not good enough.

5.       The results obtained with STRING database better to describe wider. For example, Fig.7 introduces the data for pharynx cancers. What about adenoid cystic carcinoma, breast, ovarian and lung cancers mentioned in lines 363-365?

The article is needed to be corrected more accurately.

Comments on the Quality of English Language

The English is not well-understandable: present indefinite is used instead of past indefinite or past perfect: “we analyze”, “we find” etc.

Round 2

Reviewer 2 Report

Comments and Suggestions for Authors

Dear Authors,

The article have been significantly improved.

But the references are formatted incorrectly, not in style MDPI.

Author Response

Dear Reviewer,

(a) I downloaded the 'Endnote - MDPI style'

(b) added the style to Endnote, then

(c) converted the document to the MDPI reference style